# Comparing Algorithms for Estimation of Aboveground Biomass in *Pinus yunnanensis*

**Tianbao Huang** [1,2,3], **Guanglong Ou** [3], **Hui Xu** [3], **Xiaoli Zhang** [3], **Yong Wu** [3], **Zihao Liu** [3], **Fuyan Zou** [1,2], **Chen Zhang** [1,2] **and Can Xu** [1,2,*]

1. Kunming General Survey of Natural Resources, China Geological Survey, Kunming 650111, China; huangtianbao@swfu.edu.cn (T.H.); zoufuyan@mail.cgs.gov.cn (F.Z.); zhangchen002@mail.cgs.gov.cn (C.Z.)
2. Technology Innovation Center for Natural Ecosystem Carbon Sink, Ministry of Natural Resources, Kunming 650111, China
3. Key Laboratory of Southwest Mountain Forest Resources Conservation and Utilization, Ministry of Education, Southwest Forestry University, Kunming 650233, China; olg2007621@swfu.edu.cn (G.O.); swfc213@126.com (H.X.); karojan@swfu.edu.cn (X.Z.); yongwu@swfu.edu.cn (Y.W.); liuzihao@swfu.edu.cn (Z.L.)
* Correspondence: xucan@mail.cgs.gov.cn

**Abstract:** Comparing algorithms are crucial for enhancing the accuracy of remote sensing estimations of forest biomass in regions with high heterogeneity. Herein, Sentinel 2A, Sentinel 1A, Landsat 8 OLI, and Digital Elevation Model (DEM) were selected as data sources. A total of 12 algorithms, including 7 types of learners, were utilized for estimating the aboveground biomass (AGB) of *Pinus yunnanensis* forest. The results showed that: (1) The optimal algorithm (Extreme Gradient Boosting, XGBoost) was selected as the meta-model (referred to as XGBoost-stacking) of the stacking ensemble algorithm, which integrated 11 other algorithms. The $R^2$ value was improved by 0.12 up to 0.61, and RMSE was decreased by 4.53 Mg/ha down to 39.34 Mg/ha compared to the XGBoost. All algorithms consistently showed severe underestimation of AGB in the *Pinus yunnanensis* forest of Yunnan Province when AGB exceeded 100 Mg/ha. (2) XGBoost-Stacking, XGBoost, BRNN (Bayesian Regularized Neural Network), RF (Random Forest), and QRF (Quantile Random Forest) have good sensitivity to forest AGB. QRNN (Quantile Regression Neural Network), GP (Gaussian Process), and EN (Elastic Network) have more outlier data and their robustness was poor. SVM-RBF (Radial Basis Function Kernel Support Vector Machine), k-NN (K Nearest Neighbors), and SGB (Stochastic Gradient Boosting) algorithms have good robustness, but their sensitivity was poor, and QRF algorithms and BRNN algorithm can estimate low values with higher accuracy. In conclusion, the XGBoost-stacking, XGBoost, and BRNN algorithms have shown promising application prospects in remote sensing estimation of forest biomass. This study could provide a reference for selecting the suitable algorithm for forest AGB estimation.

**Keywords:** forest aboveground biomass; machine learning algorithm; remote sensing; *Pinus yunnanensis* forest; stacking ensemble; Yunnan Province; China





## 1. Introduction

As a crucial quantitative and qualitative indicator of forest ecosystems, forest biomass holds great importance in swiftly acquiring information on the quantity of biomass within forests [1,2]. Nevertheless, traditional remote sensing methods for estimating forest biomass suffer from disadvantages, such as low efficiency, high cost, and ecological damage. As a result, there is a growing interest in biomass estimation based on remote sensing methods to overcome the shortage above. Meanwhile, to enhance the accuracy of remote sensing estimation of forest aboveground biomass (AGB), researchers have conducted extensive studies utilizing diverse image data sources and algorithms [3,4].

In AGB estimation of remote sensing, the data sources typically include lidar data, multispectral data, synthetic aperture radar (SAR), hyperspectral data, and others. While lidar data is not affected by data saturation effects and exhibits high estimation accuracy, its widespread application is limited due to its high cost [4]. For free data source images, Landsat 8 OLI and Sentinel 2A multispectral images have been widely used in forest AGB remote sensing estimation due to their advantages, such as wide coverage, free acquisition, high spatial-temporal resolution, mature technology, sensitivity to SWIR band, and red-edge bands, etc., which help to monitor vegetation leaf characteristics [5–9]. In addition, the free open-access Sentinel 1A SAR, a long-wave active sensor with the advantages of day and night operation, no rain and cloud interference, senses forest geometry better than passive optical sensors and provides valuable data for mapping forest AGB [10]. Numerous studies have also shown that collaborative estimation of forest AGB using multi-source remote sensing imagery could improve estimation accuracy, especially in regions with high heterogeneity [11,12].

AGB estimation of remote sensing had many uncertainties caused by remote sensing data sources, prediction models, forest physical environment, mixed pixel, sample biomass calculation error, sampling error, image time mismatch, and other influencing factors, which limited the accuracy of remote sensing estimation of AGB [13,14]. Among these, the model plays a crucial role in the AGB estimation of remote sensing, and the selection and performance of the model directly affect the accuracy and reliability of the AGB estimate [4,15], so it was important to select a suitable algorithm for the AGB model to improve the accuracy of remote sensing estimation of AGB. Machine learning algorithms have been widely used in the estimation of forest AGB by Remote Sensing because they can capture complex non-linear relationships between variables in multiple data sources and have high estimation accuracy [16,17]. Simultaneously, the seven types of learners algorithms, such as Bagging Learners, Boosting Learners, neural networks, linear-based learners, kernel-based learners, K-nearest neighbor learners, and stacking ensemble algorithms, have been gradually used for the estimation of forest AGB by Remote Sensing.

Review of the application of seven types of learners algorithms mentioned above that are used in AGB estimation of remote sensing, the commonly used kNN algorithm has been widely applied due to its simplicity and good estimation performance [18,19]. The Random Forest (RF) algorithm has been generally recognized as an excellent choice for bagging learners due to its robustness and high accuracy, establishing it as the most commonly employed algorithm [20]. However, it is important to note that the utilization of Quantile Random Forest (QRF), an enhanced variant of the Random Forest algorithm, has relatively few applications in estimating forest AGB by remote sensing [21,22]. Extreme gradient boosting (XGBoost), an advanced tree boosting system and an enhancement of gradient boosting based on Boosting Learners, has demonstrated outstanding performance in estimating AGB from remote sensing data [23–25]. Furthermore, Guneralp et al. (2014) [26] have shown that Stochastic Gradient Boosting (SGB) outperforms other algorithms, including MARS (Multivariate Adaptive Regression Splines) and Cubist (Cubist), in the forest AGB estimation of remote sensing within the context of Boosting Learners. The Quasi-Recurrent Neural Network (QRNN) algorithm, as a part of Neural Networks Learners, has started to gain attention in the field of estimation of forest AGB by Remote Sensing. Li et al. (2023) [27] have demonstrated the capability of QRNN to effectively enhance forest AGB estimation using remote sensing data. Furthermore, Bayesian Regularized Neural Networks (BRNN) have addressed the challenges of overfitting and robustness that are commonly associated with artificial neural networks. Although BRNN has attracted considerable interest in other domains, its application in remote sensing for forest AGB estimation remains relatively unexplored [28,29]. Among linear-based learners, Alvarez-Mendoza et al. (2022) [30] have demonstrated the favorable estimation performance of Bayesian Ridge Regression (BRR) in estimating grassland forest AGB through remote sensing. Moreover, the Elastic Network Program (EN) represents a regularized version of linear regression that combines the characteristics of Tikhonov Regularization (ridge) regression and Least Absolute Shrinkage

and Selection Operator (LASSO) regression [31]. By incorporating the properties of both algorithms, it produces estimates that can be interpreted as Bayesian posterior modes under a prior distribution implied by the elastic network form. Despite the potential benefits, there have been limited studies employing remote sensing for AGB using the EN [31]. Kernel-based learners referred to a family of algorithms that utilized kernel functions to project low-dimensional data into a higher-dimensional space, enabling linear separability. These algorithms were capable of handling nonlinear problems while still relying on linear algebra [32]. Ghosh et al. (2023) [33] demonstrated that the Gaussian Process algorithm, as a kernel-based learner, enhanced the accuracy of the estimation of forest AGB by remote sensing. Additionally, Support Vector Machine (SVM), a popular machine learning technique, was found to have widespread application in the estimation of forest AGB using remote sensing resources [4]. SVM employed a kernel function, such as the radial basis function (RBF), to process nonlinear data by mapping it into a higher-dimensional feature space. The SVM with RBF Kernel algorithm (SVM–RBF) excelled at modeling nonlinear relationships between input and output variables and exhibited robustness against noise and outliers [34,35].

Moreover, integration algorithms have been shown to exhibit higher estimation accuracy compared to single algorithms, such as the Stacking ensemble algorithm, as one of the classic integration algorithms, combines the strengths of various models and has been increasingly applied in the estimation of AGB using remote sensing resources [36,37]. In Stacking ensemble models, the diversification of the base models plays a vital role in enhancing the integrated model [38]. Additionally, selecting models with high generalization capability in the second layer as meta-models can effectively address and rectify any bias present in the first-layer base learners towards the training data. The data generated in the first layer for secondary prediction can further enhance the performance of the first layer. Therefore, the model selection in the second layer holds significant importance status [39]. By comparing multiple algorithms, selecting the optimal algorithm for remote sensing estimation of AGB became an important pathway to improve the accuracy of AGB estimation of remote sensing [16,38,40].

Although most of the different algorithms have been used in AGB estimation by remote sensing, there are still incomplete comparisons of algorithms for different learners, and some algorithms have not been investigated in forest AGB estimation, especially in highly heterogeneous landscapes. Yunnan Province is located in a longitudinal ridge and valley area with complex geological conditions and a special geographical location, resulting in high forest heterogeneity [41–43]. Accurately estimating forest AGB using remote sensing in such areas is undoubtedly a challenge [17]. For this reason, this study selected Sentinel 2A, Sentinel 1A, Landsat 8 OLI, and Digital Elevation Model (DEM) as data sources and selected 12 algorithms that pairs of bagging learners, boosting learners, neural networks, linear-based learners, kernel-based learners, KNN and stacking ensemble learners to explore the remote sensing estimation of *Pinus yunnanensis* forests in Yunnan Province.

The aims of this study were: Comparing the performance of 12 algorithms from 7 types of learners on AGB estimation of *Pinus yunnanensis* forests in highly heterogeneous landscapes.

## 2. Study Area and Materials

### 2.1. Study Area

Yunnan Province is located between 97°31′–106°11′ E and 21°8′–29°15′ N, on the Yunnan-Guizhou Plateau, predominantly mountainous and highland, with a total area of approximately 394,000 square kilometers in southwestern China, bordering the southeastern edge of the Tibetan Plateau [41,44]. The terrain slopes from northwest to southeast, with an altitude of 74–6457 m. Yunnan has a highland tropical monsoon climate with average summer and winter temperatures of 19–22 °C and 6–8 °C, respectively. *Pinus yunnanensis* is an endemic species of southwestern China. It generally grows in the plateau mountains and medium-high valleys at an altitude of 250–3500 m and was concentrated at an altitude of 1600–2900 m. The main dominant tree species in Yunnan Province are *Pinus Yunnanensis*,

*Pinus kesiya*, *Pinus armindii*, oaks, *Alnus nepalensis* and other tree species. The *Pinus yunnanensis* was the forest type with the largest distribution area in Yunnan Province, and its horizontal distribution extended to 28°23′33″ N in the north, 23°01′20″ N in the south, 97°46′39″ E in the east and 105°54′05″ E in the west [45]. *Pinus yunnanensis* not only plays an important role in the ecological benefits of soil and water conservation in plateau areas but also brings high economic and social benefits [45]. Figure 1 shows the location of the study area.

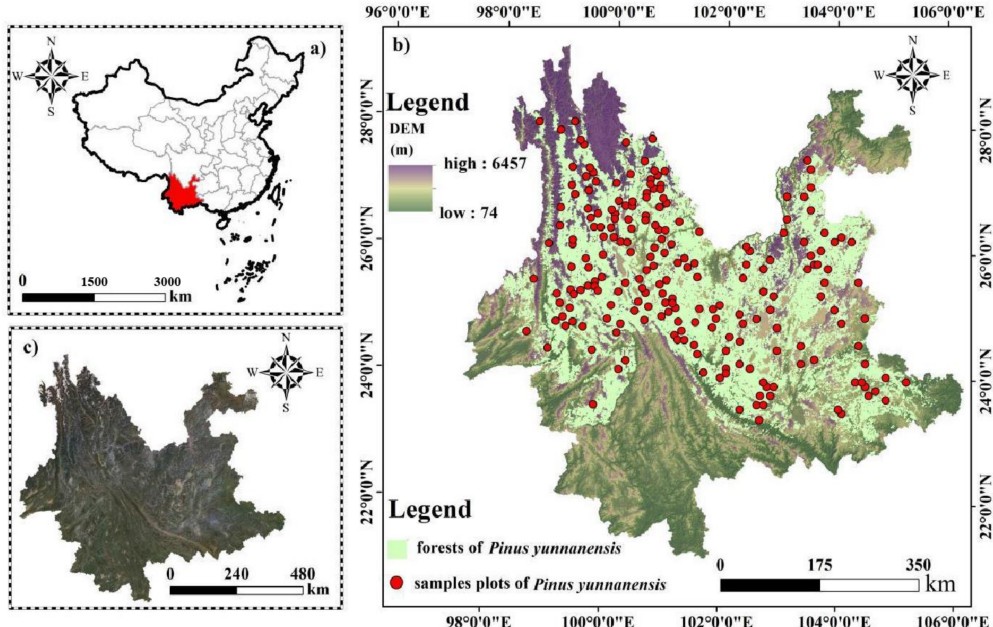

**Figure 1.** Location of the study area: (**a**) the location of Yunnan province in China, (**b**) sample plots of *Pinus yunnanensis* in Yunnan province, (**c**) the Landsat 8 OLI image.

*2.2. AGB Calculation of Pinus yunnanensis Sample Plots*

The ground data were derived from the 2021 survey of 210 *Pinus yunnanensis* forest plots in the Continuous Forest Inventory (CFI) of Yunnan Province, and the distribution of sample plots is shown in Figure 1. The basic information, such as the dominant species, the diameter at breast height (*DBH*) of individual trees, tree height, average height and the coordinate, and plot coordinates, was recorded by terrestrial RTK. The survey accuracy met the requirements of the Technical Regulations for Continuous Inventory of Forest Resources. Calculation of the aboveground biomass of individual *Pinus yunnanensis* trees is based on Liu et al. (2015), and the $R^2$ was 0.99 in equation [46], the equation was:

$$M = 0.048 \times DBH^{1.9276} \times H^{0.9638} \tag{1}$$

where *DBH* (cm) is the average diameter at breast height (1.3 m), *H* (m) is the average tree height, and *M* is the aboveground biomass of a single standing tree (kg).

To obtain the AGB of the sample plot, the unit was converted into the value per hectare using Equation (2). The final *AGB* statistical data of the *Pinus yunnanensis* forests are shown in

$$AGB = \frac{n \times M}{25.8 \times 25.8} \times \frac{10,000}{1000} \tag{2}$$

where *M* was the aboveground biomass of a single standing tree (kg), *n* was the number of trees in the sample plot, and *AGB* was the AGB of the sample plot (Mg/ha). Seventy percent modeling and 30 percent model evaluation were adopted. The sample basic information is shown in Figure 2. The 147 plots were used for model construction, and 63 plots were used for model evaluation, and there was little difference between the model sample and the test sample (*p* = 0.94 for Wilcoxon test).

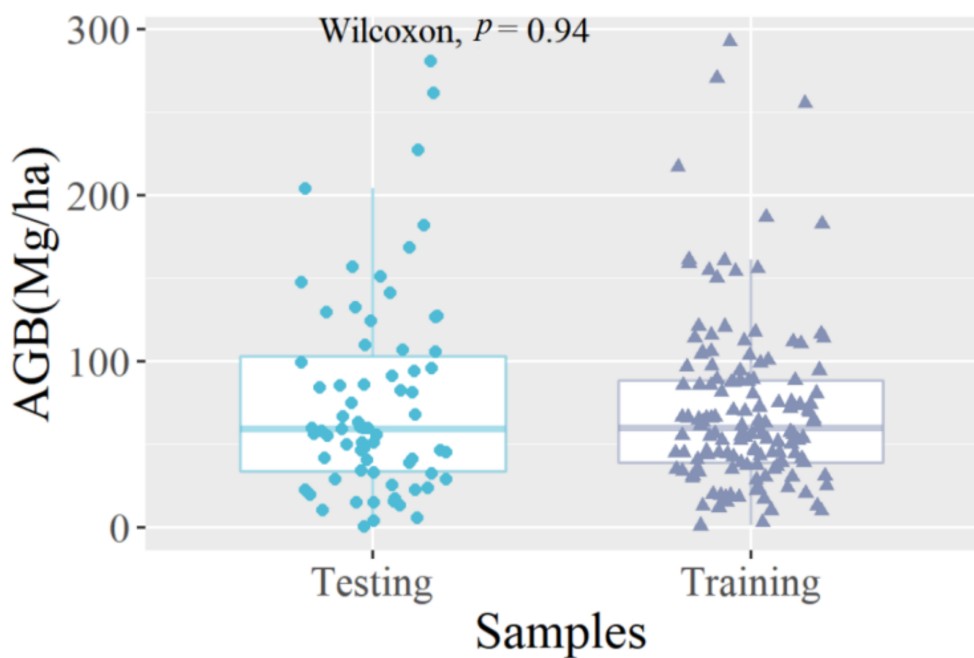

**Figure 2.** Basic overview of the samples.

*2.3. Remote Sensing Data Acquisition and Variable Extraction*

The DEM data was from the Geospatial Data Cloud (http://www.gscloud.cn/ accessed on 9 July 2023) at a spatial resolution of $30 \times 30$ m (obtained by space-borne sensors). Sentinel 1A, Sentinel 2A, and Landsat 8 OLI were downloaded from Google Earth Engine (https://code.earthengine.google.com/ accessed on 9 July 2023) to match the survey data. Sentinel 2A and Landsat 8 OLI data were surface reflectance products that selected less than 3% cloud shadow and 5% cloud to synthesize from median values of the Yunnan area in 2021 January–December. The Landsat 8 OLI was from "LANDSAT/LC08/C01/T1_SR", and Sentinel 2A was from "COPERNICUS/S2_SR" in Google Earth Engines. Sentinel 1A was from "COPERNICUS/S1_GRD" in Google Earth Engines. The Sentinel-1 mission provides data from a dual-polarization C-band Synthetic Aperture Radar (SAR) instrument at 5.405 GHz (C band). This collection includes the S1 Ground Range Detected (GRD) scenes, processed using the Sentinel-1 Toolbox to generate a calibrated, ortho-corrected product. Besides, the image synthesis time was on 20 January 2023 and resampled by $30 \times 30$ m. Subsequently, a 30 m resolution DEM was used for the terrain correction of Sentinel 2A, Landsat 8 OLI, and Sentinel 1A. The vegetation indices, single band, and texture features of Sentinel 2A and Landsat 8 OLI were calculated in ENVI 5.3 [47,48]. Landsat 8 OLI included 7 spectral bands, 17 vegetation indices, and 168 texture variables ($3 \times 3$, $5 \times 5$, $7 \times 7$ were from the gray-level co-occurrence matrix GLCM). Sentinel 2A included 12 spectral bands, 18 vegetation indices, and 168 texture variables ($3 \times 3$, $5 \times 5$, $7 \times 7$ grey-scale co-occurrence matrix feature GLCM). The spectral variables are shown in Table 1.

**Table 1.** Spectral variables.

| Image Source | Index | Abbreviation | Formula |
|---|---|---|---|
| Sentinel 1A | vertical transmit-vertical channel | VV | - |
| | vertical transmit-horizontal channel | VH | - |

Table 1. *Cont.*

| Image Source | Index | Abbreviation | Formula |
|---|---|---|---|
| Sentinel 2A | B2-Blue, B3-Green, B4-Ged, B5-Gegetation red edge, B6-Vegetation red edge, B7-Vegetation red edge, B8-NIR, B9-Water vapour, B10-SWIR-Cirrus, B11-SWIR, | B2, B3, B4, B5, B6, B7, B8, B9, B10 | - |
| | ratio vegetation index | RVI | B8/B4 |
| | difference vegetation index | DVI | B8 − B4 |
| | weighted difference vegetation index | WDVI | B8 − 0.5 × B4 |
| | infrared vegetation index | IPVI | B8/(B8 + B4) |
| | perpendicular vegetation index | PVI | $\sin(45) \times B8 - \cos(45) \times B4$ |
| | normalized difference vegetation index | NDVI | (B8 − B4)/(B8 + B4) |
| | NDVI with band4 and band5 | NDVI45 | (B5 − B4)/(B5 + B4) |
| | NDVI of green band | GNDVI | (B7 − B3)/(B7 + B3) |
| | inverted red edge chlorophyll index | IRECI | (B7 − B4)/(B5/B6) |
| | soil adjusted vegetation index | SAVI | 1.5 × (B8 − B4)/ 8 × (B8 + B4 + 0.5) |
| | transformed soil-adjusted vegetation index | TSAVI | 0.5 × (B8 − 0.5 × B4 − 0.5)/(0.5 × B8 + B4 − 0.15) |
| | modified soil-adjusted vegetation index | MSAVI | (2 − NDVI × WDVI) × (B8 − B4)/8 × (B8 + B4 + 1 − NDVI × WDVI) |
| | sentinel-2 red edge position index | S2REP | 705 + 35 × [(B4 + B7)/ 2 − B5] × (B6 − B5) |
| | red edge infection point index | REIP | 700 + 40 × [(B4 + B7)/ 2 − B5]/(B6 − B5) |
| | atmospherically resistant vegetation index | ARVI | B8 − (2 × B4 − B2)/ B8 + (2 × B4 − B2) |
| | pigment-specific simple ratio chlorophyll index | PSSRa | B7/B4 |
| | meris terrestrial chlorophyll index | MTCI | (B6 − B5)/(B5 − B4) |
| | modified chlorophyll absorption ratio index | MCARI | [(B5 − B4) − 0.2 × (B5 − B3)] × (B5 − B4) |
| Landsat 8 OLI | band1—coastal aerosol, band2—blue (BLU), band3—green (GRN), band4—red (RED), band5—near-infrared (NIR), band6—shortwave infrared 1 (SWIR1), and band7—shortwave infrared 2 (SWIR2). | B1, B2, B3, B4, B5, B6, B7 | - |
| | normalized difference vegetation index | NDVI | (B5 − B4)/(B5 + B4) |
| | NDVI with band3 and band4 | ND43 | (B4 − B3)/(B4 + B3) |
| | NDVI with band6 and band7 | ND67 | (B6 − B7)/(B6 + B7) |
| | NDVI with band3 and band5 with band6 | ND563 | ((B5 + B6) − B3)/ (B5 + B6 + B3) |

**Table 1.** *Cont.*

| Image Source | Index | Abbreviation | Formula |
|---|---|---|---|
| | difference vegetation index | DVI | B5 − B4 |
| | soil-adjusted vegetation index | SAVI | $((1 + 0.5) \times (B5 - B4))/(0.5 + B5 + B4)$ |
| | ratio vegetation index | RVI | B4/B3 |
| | brightness vegetation Index | B | $0.2909 \times B2 + 0.2493 \times B3 + 0.4806 \times B4 + 0.5568 \times B5 + 0.4438 \times B6 + 0.1706 \times B7$ |
| | greenness vegetation Index | G | $-0.2728 \times B2 - 0.2174 \times B3 - 0.5508 \times B4 + 0.7221 \times B5 + 0.0733 \times B6 - 0.1648 \times B7$ |
| | temperature vegetation index | W | $0.1446 \times B2 + 0.1761 \times B3 + 0.3322 \times B4 + 0.3396 \times B5 - 0.6210 \times B6 - 0.4186 \times B7$ |
| | atmospherically resistant vegetation index | ARVI | $(B5 - (2 \times B4 - B2))/(B5 + (2 \times B4 - B2))$ |
| | mid-infrared temperature vegetation index | MV17 | $(B5 - B7)/(B5 + B7)$ |
| | modified soil adjusted vegetation index | MSAVI | $(2 \times B5 + 0.25 - ((2 \times B5 + 0.25)^2 - 8 \times (B5 - B4))^{0.5})/2$ |
| | multiband Linear combination of band2 with band3 and band4 | VIS234 | B2 + B3 + B4 |
| | multiband Linear combination | ALBEDO | B2 + B3 + B4 + B5 + B6 + B7 |
| | Simple Ratio Index | SR | B5/B4 |
| | improved vegetation index | SAV12 | $B5 + 0.5 - ((B5 + 0.5)^2 - 2 \times (B5 - B4))^{0.5}$ |
| | optimized Simple Ratio vegetation Index | MSR | $(B5/B4 - 1)/(B5/B4 + 1)^{0.5}$ |
| | karst terrain factor 1 | KT1 | $0.304 \times B2 + 0.279 \times B3 + 0.474 \times B4 + 0.559 \times B5 + 0.508 \times B6 + 0.186 \times B7$ |
| | principal component 1—factor A | PC1-A | $0.054 \times B2 + 0.130 \times B3 + 0.143 \times B4 + 0.595 \times B5 + 0.709 \times B6 + 0.321 \times B7$ |
| | principal component 1—factor B | PC1-B | $0.140 \times B2 + 0.242 \times B3 + 0.313 \times B4 + 0.262 \times B5 + 0.739 \times B6 + 0.457 \times B7$ |
| | principal component 1—factor P | PC1-P | $0.056 \times B2 + 0.079 \times B3 + 0.127 \times B4 - 0.845 \times B5 - 0.490 \times B6 - 0.143 \times B7$ |

## 3. Research Method

In this study, Sentinel 1A, Sentinel 2A, Landsat 8 OLI, and DEM were utilized as data sources. The research was conducted based on 210 FCI *Pinus yunnanensis* forest plots in Yunnan Province. A total of 12 algorithms from 7 types, including Bagging learners, Boosting learners, neural network, linear-based learners, kernel-based learners, kNN, and Stacking ensemble, were constructed for the study. Among these algorithms, the Stacking ensemble algorithm integrates 11 algorithms, namely Bagging learner, Boosting learner,

neural network, linear learner, kernel function learner, and kNN, from 6 types of learners. The Stacking ensemble then selected the optimal algorithm from these 6 types of learners as its meta-model. The research workflow is illustrated in Figure 3.

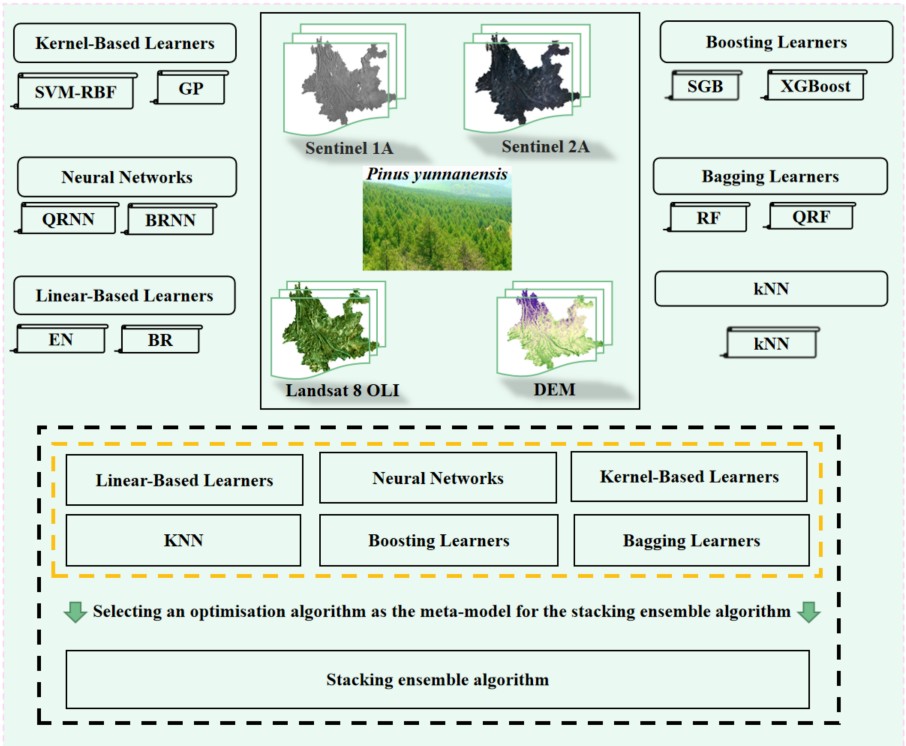

**Figure 3.** Research technical route.

## 3.1. Variable Selection

Linear stepwise regression (LSR) is a commonly employed variable selection method in remote sensing-based estimation of forest AGB [20,48]. It aims to identify useful variables from a pool of redundant data. LSR introduces characteristic variables into the model and conducts significance tests one by one to identify statistically significant variables that fall within a specified range ($p < 0.05$). These selected variables form the final combination in LSR [48]. To ensure the accuracy of the estimation model, collinearity between the selected trait variables is assessed for each variable combination. Collinearity refers to the presence of strong correlations between predictor variables, which can introduce bias in the model estimation. In this context, the variance inflation factor (VIF) is used as a measure of collinearity between the trait variables. A VIF threshold of 10 is commonly applied to detect and address collinearity issues [38,49]. Variables with VIF values exceeding this threshold are considered to have high collinearity and may be excluded from the model to mitigate bias.

## 3.2. Model Construction

Grid search is a common tuning method that optimizes model performance by searching for the best combination of hyperparameters in a predefined hyperparameter grid [50]. This research adopts 5 cross-validation search algorithms to find the optimal combination of parameters, parameters CARET used to wrap the default value (https://topepo.github.io/caret/ accessed on 9 July 2023). The hyperparameters of the algorithm are shown in Table 2.

**Table 2.** Algorithms for tuning hyperparameters.

| Algorithm | R Packages | Hyperparameters Tuned |
|---|---|---|
| Random Forest | Random Forest | mtry (Randomly Selected Predictors) |
| Quantile Random Forest | Quantreg Forest | mtry (Randomly Selected Predictors) |
| Gaussian Process | kernlab | none |
| Stochastic Gradient Boosting | gbm, plyr | n.trees (Boosting Iterations), interaction.depth (Max Tree Depth), shrinkage (Shrinkage), n.minobsinnode (Min. Terminal Node Size) |
| Support Vector Machines with Radial Basis Function Kernel | kernlab | Sigma (Sigma), C (Cost) |
| Bayesian Regularized Neural Networks | brnn | neurons |
| Quantile Regression Neural Network | qrnn | n.hidden (Hidden Units), penalty (Weight Decay), bag (Bagged Models) |
| Bayesian Ridge Regression | monomvn | None |
| Gaussian Process | kernlab | None |
| Elasticnet | elasticnet | fraction (Fraction of Full Solution), lambda (Weight Decay) |
| K-nearest neighbor | none | k (Neighbors) |
| Extreme gradient boosting | xgboost, plyr | nrounds, max_depth, eta, gamma, subsample, colsample_bytree, rate_drop, skip_drop, min_child_weight |
| Stacking ensemble | caretEnsemble, mlbench, caret | - |

notes: "-", the hyperparameters are determined by the meta-model of Stacking ensemble algorithm.

### 3.3. Model Evaluation

Using the sample independence test to calculate its coefficient of determination ($R^2$) and root mean square error (RMSE) metrics for model evaluation.

$$R^2 = 1 - \frac{\sum_{i=1}^{n}(y_i - \hat{y}_i)^2}{\sum_{i=1}^{n}(y_i - \overline{y}_i)^2} \tag{3}$$

$$RMSE = \sqrt{\frac{\sum_{i=1}^{n}(\hat{y}_i - y_i)^2}{n}} \tag{4}$$

where $n$ is the number of sample observations, and $y_i$ is the actual value; $\hat{y}_i$ is the estimated value, and $\overline{y}_i$ is the mean of the observed sample.

## 4. Results

### 4.1. Variable Selection

A total of 224 feature variables (35 vegetation indices, 19 single bands, 168 texture features, VV, VH) were selected using the LSR method. Figure 4 shows the selection results, and it can be seen that there was no strong multi-collinearity between variables. Eight variables were selected to participate in the model construction, including three vegetation indices (S2PSSRa, S2NDVI45, S2REP), four texture features (L8_b7_EN7, L8_b5_CR7, L8_b6_SM5, L8_b3_CR5) and one terrain factor (Slop). The results showed S2PSSRa had the highest correlation with forest AGB. The figure also showed that the biomass samples were skewed to the right.

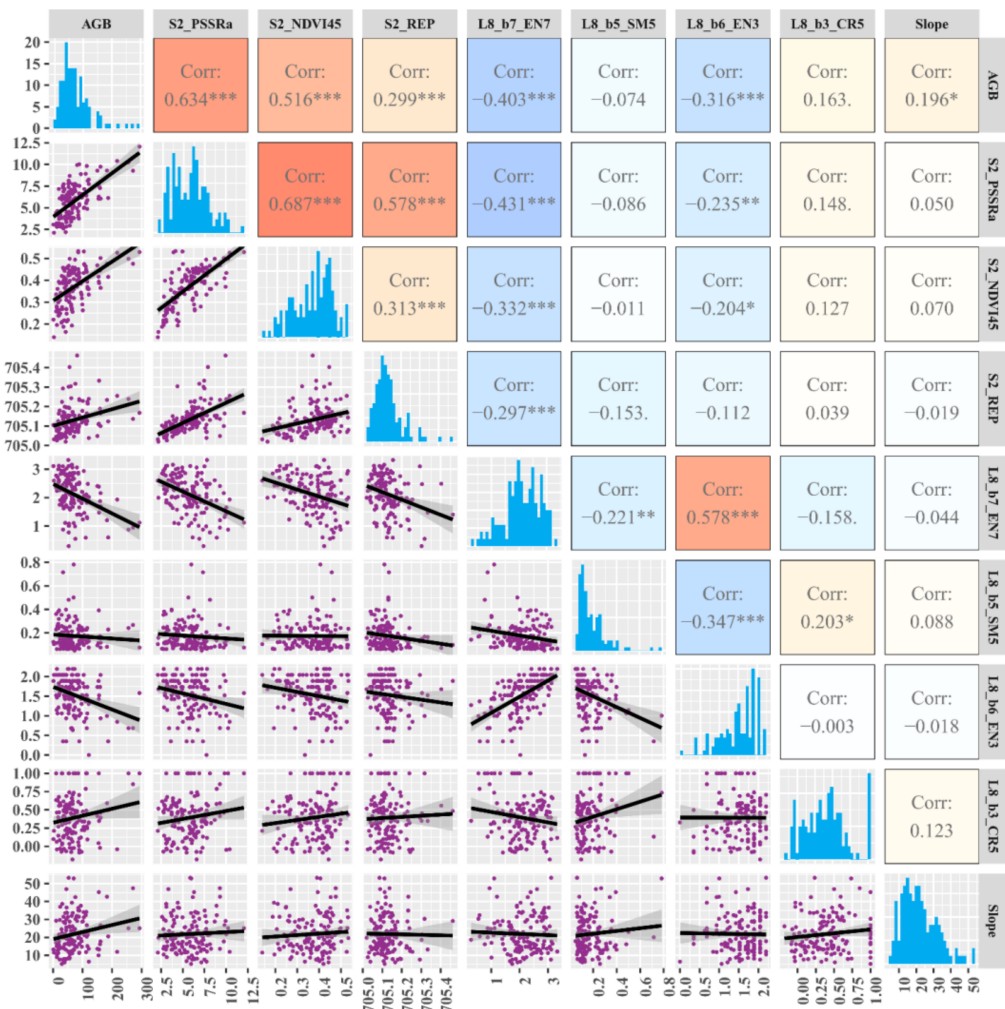

**Figure 4.** Correlation of variables (the blue diagonal line is the data distribution frequency map, the lower left corner is the scatterplot relationship between the two variables, and the upper right corner is the Pearson correlation coefficient between the two variables. Note: "*" stands for $p \leq 0.05$, "**" stands for $p \leq 0.01$, "***" stands for $p \leq 0.001$).

### 4.2. Model Evaluation

Figures 5 and 6 were the fitting diagrams and AGB maps of the models, respectively. Figure 5 shows that the goodness of fit at a single algorithm level was XGBoost > BRNN > EN > GP > RF > QRF > BR > QRNN > SGB > SVM-RBF > kNN. In addition, it can be seen from the AGB maps that the distribution of high and low AGB values was almost consistent across all algorithms. XGBoost, BRNN, RF, and QRF had good sensitivity and the range of AGB estimates was reasonable with good robustness, among which QRNN, GP, and EN had more outliers and poor robustness. The SVM-RBF, k-NN, and SGB also had good robustness, but their sensitivity was less than that of XGBoot, BRNN, RF, and QRF algorithms, and high-value underestimation was evident. For this purpose, the XGBoot algorithm was chosen as a meta-model of the Stacking ensemble to integrate 11 algorithms. It can be seen that the stacking integrated algorithm has the highest estimation accuracy ($R^2 = 0.61$, RMSE = 39.34 Mg/ha). Based on the XGBoost-stacking algorithm integrated with 11 algorithms, $R^2$ increased by 0.12 and RMSE decreased by 4.53 Mg/ha compared to the single optimal algorithm, XGBoost algorithm. Good sensitivity and robustness were also reflected in the AGB maps of the XGBoost-stacking algorithm. However, it can be seen from Figure 5 that although the integrated stacking algorithm mitigated high-value underestimation to some extent, all algorithms showed high-value underestimation in forest AGB at around 100 Mg/ha, especially the SVM-RBF algorithm. However, for low

values, the BRNN algorithm was more practical and could estimate low AGB values. In conclusion, XGBoost, BRNN, and XGBoost-stacking algorithms have a good application prospect in AGB estimation of remote sensing, and high-value underestimation was still an important factor affecting the accuracy of AGB estimation.

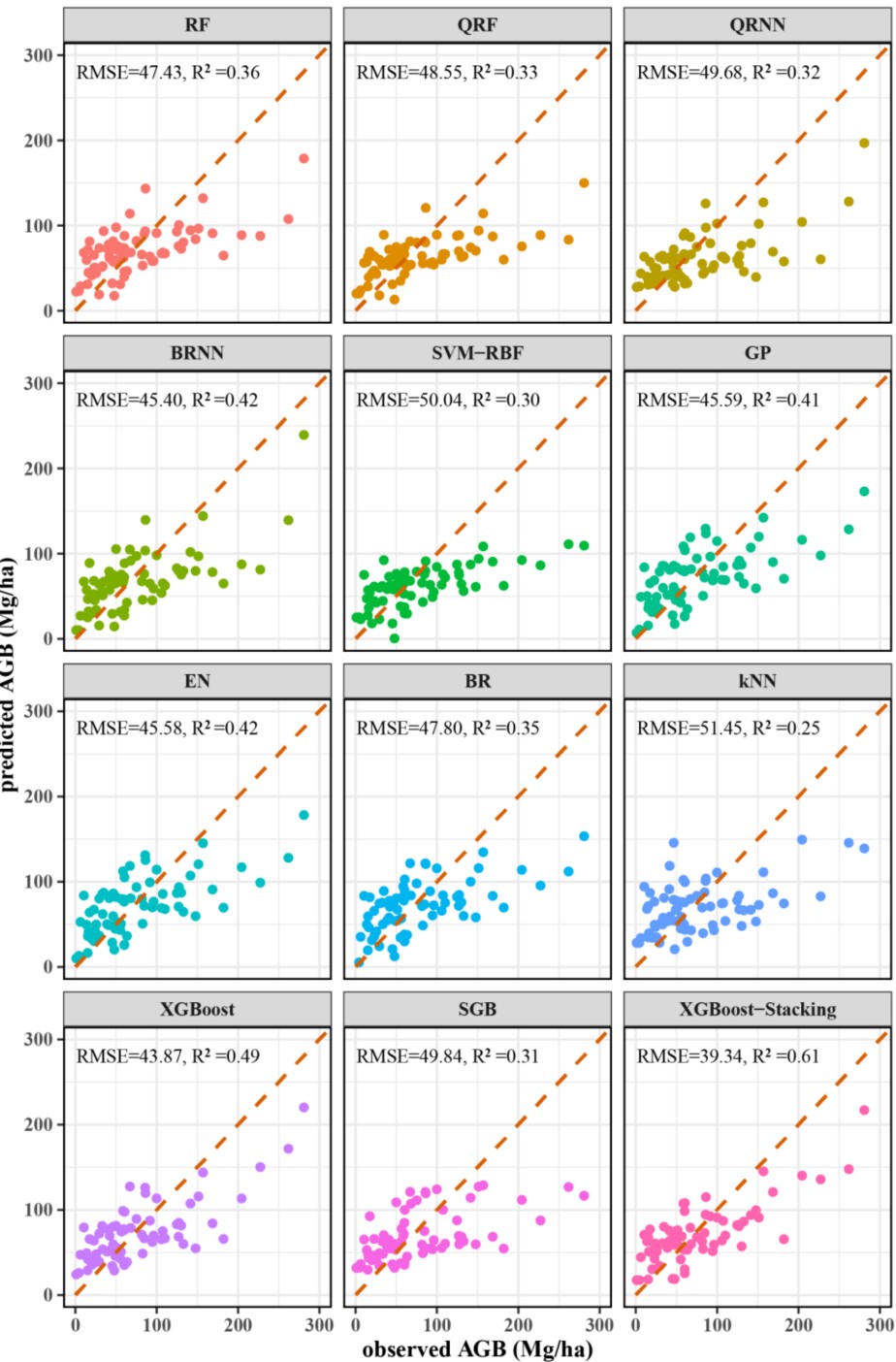

**Figure 5.** AGB fitting scatter diagram of 12 algorithms.

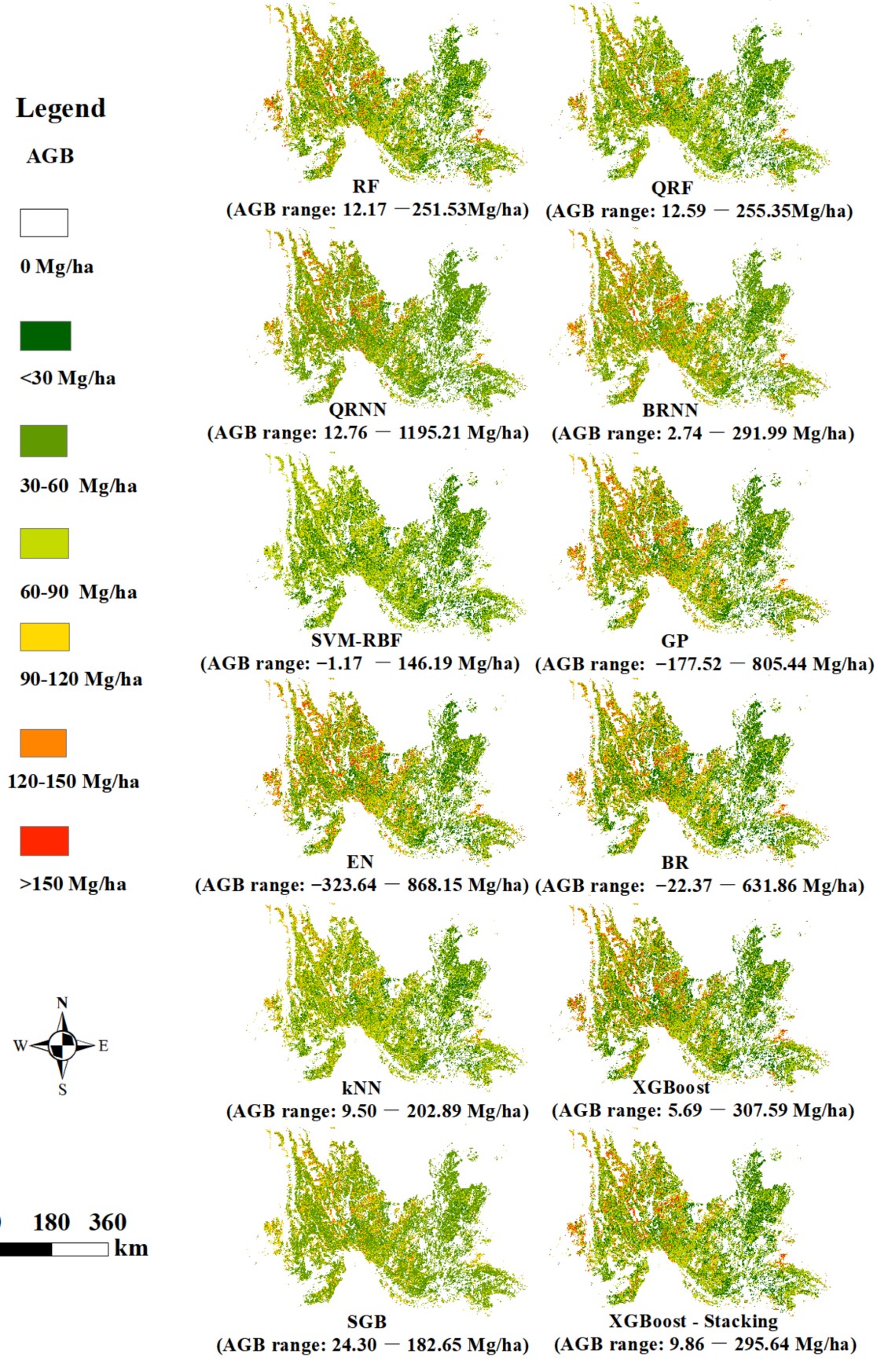

**Figure 6.** AGB maps of 12 algorithms in study areas.

## 5. Discussion

Theoretically, Sentinel 1A data have a strong ability to penetrate forest stands and may well reflect the vertical structure of forest stands, which correlates well with AGB [4]. However, in this study, the backscattering coefficient of Sentinel 1A was not selected by LSR variables to participate in the model construction. This may be because the correlation between the backscatter coefficient of SAR and the AGB of the forest is easily affected by complex terrain. High-precision terrain correction for SAR images can improve image quality, and the accuracy of the DEM greatly affects image quality [51–54]. For example, in Vatandaşlar's research, 1-m resolution data from the Shuttle Radar Topography Mission (SRTM) was used to perform terrain correction on SAR data, and a good estimation effect was obtained in mountainous landscapes [55]. However, *Pinus yunnanensis* in Yunnan Province generally grows in plateau mountains and mid-altitude valleys at an elevation of 250–3500 m, and the terrain is relatively complex. In this study, 30 m DEM was used to perform terrain correction on Sentinel 1A data. The DEM data are rough, which may be the reason for the correlation between Sentinel 1A and AGB. Besides, this study employed the LSR variable selection method to identify variables with high linear correlation to forest AGB. To control collinearity between variables and mitigate estimation instability, a VIF threshold of <10 was applied. Additionally, variable selection methods, such as LASSO [56] and Boruta [17] may help prevent the exclusion of SAR data with low linear correlation coefficients from the analysis. If the variable selection process is avoided, directly combining SAR and optical variables to build a model can improve the accuracy of AGB estimation. Moreover, existing studies have demonstrated that integrating texture measurements from SAR images with forest auxiliary information can further enhance the AGB estimation in mountainous forest remote sensing [57,58], which should be improved in future research. Furthermore, from the perspective of the importance of the variables, the correlation of the vegetation index was higher than that of the texture features, and the forest structure of coniferous forests was simpler than that of deciduous forests, reflecting the higher correlation of the vegetation index in forests with simpler forest structure [4,13].

The XGBoost algorithm demonstrated excellent fitting and robustness in this study, primarily due to its inclusion of regularization techniques and pruning strategies. These components play a crucial role in controlling the complexity of the model and mitigating the risk of overfitting [59]. XGBoost has integrated the prediction results of all the basic learners. Furthermore, during the learning and storage process, XGBoost has utilized various methods to address the challenge of missing values encountered at different nodes. Additionally, XGBoost has provided support for custom loss functions and incorporated regular terms into the objective function to simplify the learning model and improve the overall learning effectiveness. As a result, the XGBoost-based algorithm has proven to be effective for estimating forest AGB. BRNN were more robust than QRNN because they can control the number of effective parameters for training through a Bayesian criterion and are insensitive to the architecture of the network [60]. It has been observed that the algorithm employing the regularization techniques had promising prospects for the estimation of forest AGB through remote sensing. Particularly in the areas characterized by high forest heterogeneity, the algorithm exhibited better robustness and superior fitting performance. In addition, the Stacking integrated algorithm had the highest estimation accuracy in this study, the $R^2$ was 0.61, and the RMSE was 39.34 Mg/ha. The estimation performance of the Stacking algorithm was largely dependent on the performance of the meta-model. This research compared the performance of 11 machine learning algorithms and selected the optimal algorithm of XGBoost as the meta-model for the Stacking ensemble. The XGBoost-stacking algorithm not only combined the excellent performance of the XGBoost algorithm and Stacking algorithm but also integrated the excellent performance of six kinds of learning algorithms of the basic model. Therefore, the XGBoost-stacking algorithm could significantly improve the remote sensing estimation performance of forest AGB by improving the generalization ability of the model [36]. Although the accuracy was improved, the increase was not large. Besides, if we can screen from the model level,

eliminate redundant model variables, and select truly useful model variables for Stacking ensemble, the estimation accuracy may be better improved. Furthermore, only the fusion strategy of stacking algorithms was considered in this study. The fusion strategies of other algorithms, such as blending ensemble learning [61] and averaging algorithms, can be explored in future research. The saturation effect was a common phenomenon in AGB estimation based on optical remote sensing [4]. The saturation phenomenon was serious according to the scatter distribution of predicted and observed values, and there was a serious underestimation for higher AGB values.

Compared with similar studies, it can be seen that the AGB of *Pinus yunnanensis* was underestimated when AGB was greater than 100 Mg/ha, which led to a lower accuracy of estimation in this study. The saturation effect threshold of the data in this study was lower than the AGB saturation of pine forests in Zhejiang Province of 159 Mg/ha, as reported by Zhao et al. (2016) [62]. This may be due to the AGB samples showing a right-skewed distribution that most of the values clustered in the low-value and fewer AGB samples with a high value, indicating a certain degree of forest heterogeneity, which was also an important reason for the serious underestimation of the higher values (Figure 4). At the same time, the structure and habitat of the forest were more complex in Yunnan Province compare to Zhejiang Province. Compared to Tang et al. (2022) and Chen et al. (2022) [41,44] remote sensing assessment of forest AGB in Yunnan Province, the remote sensing estimation accuracy of AGB in this study was still lower. If the hierarchical estimation of *Pinus yunnanensis* forest could be implemented according to the characteristics of topography and phenology, the data saturation phenomenon could be reduced, and the estimation accuracy would be improved. Due to the limitation of the sample size, stratified estimation was not possible in this study, but it could be supplemented in future studies. In addition, LiDAR and high-resolution optical remote sensing data can provide the vertical distribution information of vegetation and richer spectral characteristics, and introducing them into the remote sensing estimation of *Pinus yunnanensis* AGB may overcome and reduce the data saturation effect and improve the estimation accuracy [63,64]. Some studies showed that temperature factors had a significant influence on coniferous forests in Yunnan Province. Thus, adding environmental factors, such as temperature, would reduce the phenomenon of underestimation and overestimation [17,65,66]. In addition, the combination of geostatistical methods could be used for the next step study, as it can also reduce the spatial heterogeneity of forest images, as well as the data saturation effect, which may further improve the estimation accuracy [67].

## 6. Conclusions

Research shows that among the 12 algorithms, the fitting performance rank was XGBoost-Stacking > XGBoost > BRNN > EN > GP > RF > QRF > BR > QRNN > SGB > SVM-RBF > kNN. The stacking ensemble, with XGBoost as the meta-model, achieved the highest estimation accuracy, with an $R^2$ value of 0.61 and an RMSE of 39.34 Mg/ha. When compared to the single optimal XGBoost algorithm, the stacking ensemble showed an improvement of 0.12 in $R^2$ and a reduction of 4.53 Mg/ha in RMSE. The Stacking, XGBoost, BRNN, RF, and QRF models had a good sensitivity to AGB, which obtained a reasonable AGB estimation range and good robustness. On the contrary, the QRNN, GP, and EN models had more outlier data and poor robustness. SVM-RBF, k-NN, and SGB algorithms also had good robustness, but their sensitivity was worse than that of XGBoot, BRNN, RF, and QRF algorithms, and many of larger values were underestimated. All algorithms underestimated the values when the forest AGB > 100 Mg/ha, especially the SVM-RBF algorithm. However, for lower values, the BRNN algorithm was more practical and could estimate lower AGB with more accuracy. In conclusion, XGBoost, BRNN, and XGBoot-Stacking had a good application prospect in AGB estimation of remote sensing, and high-value underestimation was still an important factor affecting the accuracy of AGB estimation.

In the optical remote sensing-based estimation of *Pinus yunnanensis* forest AGB in highly heterogeneous areas of Yunnan Province, the saturation effect was still an important factor affecting the accuracy. XGBoot-Stacking could improve the estimation accuracy and selecting an appropriate algorithm to participate in the AGB remote sensing estimation is the key step to reducing the estimation errors. This study could provide a reference for selecting suitable algorithms and data sources in AGB estimation.

**Author Contributions:** Conceptualization: T.H., X.Z., Y.W., G.O. and C.X.; data curation: T.H. and Z.L.; formal analysis: T.H. and C.X.; funding acquisition: C.X.; investigation: F.Z. and C.Z.; methodology: T.H. and G.O.; project administration: C.X., C.Z. and F.Z.; resources: C.X., F.Z. and C.Z.; software: T.H. and Z.L.; supervision: C.X.; validation: T.H. and G.O.; visualization: T.H.; writing original draft: T.H.; writing review and editing: T.H., Y.W., X.Z. and H.X. All authors have read and agreed to the published version of the manuscript.

**Funding:** This work was supported by the comprehensive survey of carbon sinks in typical areas of China by Kunming Natural Resources Survey Center of China Geological Survey (DD20220877) and the Expert Workstation of Yunnan Province of China grant number 2018IC100).

**Data Availability Statement:** No new data is created.

**Conflicts of Interest:** The authors declare no conflict of interest.

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
