# Peer review of "Comparing Algorithms for Estimation of Aboveground Biomass in Pinus yunnanensis"

_forests, doi:10.3390/f14091742_

Round 1

Reviewer 1 Report

In general, there is an unbalance between the modeling and other parts of the manuscript. The authors focus more on the learners, while they seem to somehow skip other contents superficially, which may be also important for Forests readers.

Please find my specific comments below.  

Title

“Multi-algorithm”> m should be lowercase in the title.

Abstract

The first sentence is not clear. Should be rephrased (by types of what?, use only “by” or “using”).

DEM is not a satellite like others. Please provide information on the source of DEM data (is it space-borne or airborne?)

1 more > one more. Or state it differently.

Poor English use in the abstract compared to Intro. Must be improved.

Keywords

“aboveground biomass estimation by remote sensing” is not a keyword. Consider shortening it.

1.      Introduction

3rd paragraph: … it played a crucial role… Vague in this form. Please state “it” explicitly.

3rd paragraph: This is a very long paragraph. May be broken into smaller parts. Hard to follow. In addition, the information provided in other paragraphs seems to be superficial compared to this one. As such, other paragraphs should be improved a little bit.

Page2, Line 87: Why is the use of QRF limited in the estimation of forest AGB? Provide the specific reasons…

L106: lasso regression. It is an abbreviation. Use uppercase letters with its expansion pls.

L116-117: “forest biomass remote sensing estimation”. Weird statement. Correct pls.

L131: “remote sensing estimation of AGB”. May be stated differently. E.g., estimation of AGB using remote sensing resources.

L137: “heterogeneous landscape areas” > heterogeneous landscapes.

L140: Did you use 11 or 12 algorithms in this study? There is a mismatch with the information in the abstract. Clarify it pls.

2.      Study area

Remove the first sentence! “Interventional studies involving animals or humans, and other studies that require 147 ethical approval, must list the authority that provided approval and the corresponding 148 ethical approval code.”

L152: “down warded”. Awkward statement.

“394,000 squares” squares kilometers?

74 to 6457 m. Is this correct? It is a dramatic increase in the sea level.

Use a typical hyphen (-) between the altitude values.

Soil and water conservation is also an ecological benefit. Revise this statement, please.

What are other tree species in your study area? State them in this subsection

2.2. AGB calculation

“Them sample plots consisted of 210 plots” Correct this statement, please.

“Evenly distributed” means systematical sampling. You can remove one of them.

What’s the shape of plots? What’s the interval for your age classes (5-year, 10-year?)? What does stand condition mean? Clarify pls.

RTK can be used on different platforms. Specify the RTK they used (drone?, terrestrial?)

L174-175: Rephrase this sentence

What was the R2 of Liu et al.’s equation?

L180. Equation 4 or Equation 2?

“The sample status” weird statement. Replace it pls.

2.3. RS data acquisition

Why did you need to download the optical data in GEE?

You should show the equations of uncommon vegetation indices in Table 2 (e.g. RVI, DVI). They are not popular like NDVI, so readers may want to know their calculations. Crediting the developers of these indices by citing them would also be nice in a scientific paper (e.g., see https://doi.org/10.1080/02757259509532298 for ratio vegetation index (RVI) and others).

3.      Methodology

Fig. 3. A nice figure describing complex workflows.

3.1. Variable selection

This subsection needs some references. For example, how do you know that the LSR method is commonly used among researchers? Here, you should cite several studies estimating forest AGB using conventional regression methods as you did in the introduction section (e.g., https://doi.org/10.1016/j.rse.2011.03.020, etc). You can also highlight the novelty and robustness of your approach (i.e., ML) by comparing the results of those studies with yours in the discussion section. 

You also explain the LSR method by referring to collinearity, VIF and bias issues that may be faced during the modeling process. These are relevant and important topics but you should use some supporting resources from the statistics fields (e.g., https://doi.org/10.1007/978-1-84882-969-5).

Who does suggest a VIF threshold of 10 to address the collinearity issue? I could not find it. Please, cite the specific work here. There are many other studies suggesting different threshold values for the same purpose…

3.2. Algorithms (1-7)

I am not sure the introduction of each algorithm is necessary here. Since you already mentioned them in the intro, you may skip this part. At the end of the day, this is a forestry journal—readers can go and look at the details from a comprehensive resource you may cite.

4.      Results

L340. …LSR variables, …

L344. Slope, not slop.

Fig4. Please provide some description to the figure caption. For example, What do gray regions mean in scatter plots?

Metamodel?

Check and correct the words in Fig. 5.

Why did you name “estimation inversion maps” to Fig. 6? AGB maps would be clearer.

5.      Discussion and Conclusion

I do not agree with the authors regarding the first argument in the Discussion section. I do know some work conducted in mountainous landscapes that yielded good estimation results using the backscattering coefficients of Sentinel 1A as independent variables (see https://doi.org/10.1007/s11676-021-01363-3 among others). The authors should explain the poor correlation differently or they should further discuss this issue by carefully investigating earlier research conducted in mountainous landscapes. 

I also wonder why not the authors used the LASSO technique instead of LSR although they mentioned the LASSO in their paper. I believe it could solve the abovementioned issue related to Sentinel 1A data.

Please correct the subsection number of the conclusion section.

Please find my corrections related to English styles and typos in the main Comments text.

Reviewer 2 Report

Comment for forests-2524130

The context of this manuscript “A Multi-algorithm evaluation of aboveground biomass estimation by remote sensing for Pinus yunnanensis forest in Yunnan Province, China” follows the Special Issue "Advanced Applications in Remote Sensing and GIS to Forest Management and Planning". I review this manuscript and provide comment as follows.

1.      Title and Abstract are suitable that can reflect whole text. I only have a minor suggestion here. The end sentence “This study could provide a reference for selecting the suitable algorithm for pine forest AGB estimation.” could be omitted.

2.      From Introduction, audiences could understand the background information and significance of this study. However, the purposes of this study may not be explicit.  Therefore, I suggest using point by point to show the study purpose in the end of Introduction.

3.      Below equations, “where” should use lower case please format the notes of all equations.

4.      Figure 3 should have more descriptions in text.

5.      Figure 4 is not so clear; it should be improved.

6.      I suggest that “5. Discussion and conclusion” might separate as “5. Discussion” and “6. Conclusion”.

7.      Authors should emphasize the significance of results of this study compared to other references in Discussion chapter.

8.      Conclusion also should be improved, such as points 1 and 2 are too short but point 3 is long.

Reviewer 3 Report

The paper is interesting, but it needs some improvements.

Main remarks:

Abstract

Some typos must be corrected. Please re-read and correct them. The abstract must be reduced in its extension.

Example: "TheTo exploratione"

The rest of paper is OK. It has several corrections and improvements.

Author Response

Thanks for your valuable feedback and comments.On this basis, we have improved the manuscript as follows

(1) improved the readability of the abstract section and reduced the length of the abstract section

(2) Checked the manuscript for errors in grammar, spelling, etc.

(3) Re-checked the references

Reviewer 4 Report

Review Summary:

- Abstract: Needs improvement to be more readable.

- Introduction: Appropriate.

- Dataset: Sufficient.

- Other sections: Appropriate, may require minor corrections.

Overall Recommendation: Accept after minor revision.

Detailed Feedback:

1. Abstract: The abstract needs improvement to enhance readability. Consider rephrasing sentences for clarity and conciseness. Also, ensure that the key findings and contributions of the paper are well-highlighted in the abstract.

2. Introduction: The introduction is deemed appropriate, which is a positive aspect of the manuscript.

3. Dataset: It's good to know that the dataset provided is sufficient for the research.

4. Other Sections: The other sections of the manuscript are appropriate. However, some minor corrections may be needed. Be sure to review the entire manuscript for any errors or inconsistencies.

Overall, the paper has potential, but a revision of the abstract is crucial to improve its readability and better showcase the paper's key points. 

Reviewer 5 Report

A Review report on

A multi-algorithm evaluation of aboveground biomass estimation by remote sensing for Pinus yunnanensis Forest in Yunnan Province, China

General Overview:

The authors Huang et al. reviewed the uses of optical remote sensing sensor data in the estimation of above-ground biomass. For this, the authors selected six types of machine learners. Bagging, boosting, Neural Network, linear kernel-based, and k-nearest neighbor as well as the meta-learners. Among these types of models 11 candidate models were evaluated, authors found that extreme gradient boosting performed the best and k-nearest neighbor (KNN) performed the worst.

The authors found a 12 % increase in the coefficient of determination when the stacking ensemble model was used. Here authors used the term “Stacking integrated”, I have never heard of this term before. When Above Ground Biomass (AGB) was greater than 100 Mg/ha, the estimated was underestimated due to saturation effect. It appears saturation effect is applied in al the dataset used Sentinel (2A) and (1A), Landsat (8-OLI).

The title should be changed to highlight machine learning models. Abstract at his point does not offer context, clear objectives, and important results. Therefore, needs a complete rewrite.

Introduction: Introduction is poorly developed, it needs a ground of re-written however, most of the content can still be the same. Sentences are awkwardly jumping from one issue to another, more coherence is required. Further, the background on how forest biomass is usually measured and why optical remote sensing-based data are used for this purpose needs to be expanded in the early paragraphs.

The last paragraph in which the objective of the study should be mentioned is not clear. It is very difficult to understand what the authors were trying to do.  I tried to provide specific comments to the manuscript only to realize that there are several issues with grammar, scientific methods, incoherent writing, etc.  At this stage, the manuscript lacks scientific rigor and needs a complete rewrite from the beginning.

Methods:

Why is the first sentence even needed here? What ethical issues are to be met for the forest survey? It is out of context.

Approximately 394,000 squares of what? Square kilometers?

Line# 162:- Location map is shown in Figure 1?

Line# 169: The sample plots are evenly distributed? They don’t look evenly spaced. What do authors mean by evenly distributed?

Line# 177: Where should be where (small case)- change it to small in line # 183 as well.

Equation 1: If authors have the data, then why use the equation from Liu et al. (2015). Check if the citation follows the journal style it is Liu et. al or Liu et al.

Line# 180. It does not make sense to mention Equation 4 before Equation 2.

Line# 184 – 185. 70% modeling and 30 percent model evaluation were adopted. It does not make any sense.  

Line# 185- 186: The sample status was shown in Table 2. NO

Section 2.3: Did authors used median of calendar year (January to December)? How about the summer when the trees usually grow?

Co-occurrence matrix on rescaled LANDSAT or SENTINEL data ?

3. Research Method does not make sense.

Results:

4.1. Needs a complete rewrite.

Although in this case, the authors found the Stacking Ensemble model to be the better fit. However, this is not always the case. I don’t know why the authors did not use the fitted line. Rather use a scatter plot. Based on Figure 5. XGBoost and Stacking seem to be similar. What model was used in the stacking ensemble?

Based on methods and results discussion and conclusion also needs to be rewritten. I have never heard the term stacking integrated algorithm before, are you trying to say stacking ensemble?

 The study has the potential to be published in the Forests, however, at this stage lacks scientific merit. It should be completely rewritten from the ground up.

Scientific writing is lacking in the manuscript. Terminologies are inconsistent. such as an ensemble model or integrated model. 

Round 2

Reviewer 1 Report

The authors have addressed all of my concerns and improved the manuscript accordingly. However, I saw some typos in the revised text. Since it included many different colors due to changes, I couldn’t understand which parts were removed and which parts were new. Please double check the clean version of the final product in order to avoid any possible mistakes.

English styles and grammar have been improved in the revised version. 

Author Response

(The authors gave the same response as above.)
